# An Analytical Framework for the Investigation of Tropical Cyclone Wind Characteristics over Different Measurement Conditions

**Lixiao Li** [1,2] , **Yizhuo Zhou** [1] **, Haifeng Wang** [3] **, Haijun Zhou** [1,2] **, Xuhui He** [4] **and Teng Wu** [3,*]

[1] College of Civil and Transportation Engineering, Shenzhen University, Shenzhen 518060, China; lilixiao@szu.edu.cn (L.L.); 1810332047@email.szu.edu.cn (Y.Z.); haijun@szu.edu.cn (H.Z.)

[2] Guangdong Provincial Key Laboratory of Durability for Marine Civil Engineering, Shenzhen University, Shenzhen 518060, China

[3] Department of Civil, Structural and Environmental Engineering, University at Buffalo, State University of New York, Buffalo, NY 14203, USA; hwang48@buffalo.edu

[4] School of Civil Engineering, Central South University, Changsha 410083, China; xuhuihe@csu.edu.cn

[*] Correspondence: tengwu@buffalo.edu

**Abstract:** Wind characteristics (e.g., mean wind speed, gust factor, turbulence intensity and integral scale, etc.) are quite scattered in different measurement conditions, especially during typhoon and/or hurricane processes, which results in the structural engineer ambiguously determining the wind parameters in wind-resistant design of buildings and structures in cyclone-prone regions. In tropical cyclones (including typhoons and hurricanes), the inconsistent wind characteristics may be in part ascribed to the complex flow structure with the coexistence of both mechanical and convective turbulence in the boundary layer of tropical cyclones. Another significant contribution to the scattered wind characteristics is due to various measurement conditions (e.g., terrain exposure and height) and data processing schemes (e.g., averaging time). The removal of the inconsistency in the field-measurement system may offer a more rational comparison of measured wind data from various observation platforms, and hence facilitates a better identification scheme of the wind characteristics to guide the urban planning design and wind-resistant design of buildings and structures. In this study, an analytical framework was firstly proposed to eliminate the potential observation-related effects in wind characteristics and then the wind characteristics of seven field measured tropical cyclones (four typhoons and three hurricanes) were comparatively investigated. Specifically, field measurements of wind characteristics were converted to a standard reference station with a roughness length of 0.03 m, observation duration of 10 min for mean wind and averaging time of 3 s for gusty wind at a 10 m height. The differences of the measured wind characteristics between the typhoons and hurricanes were highlighted. The standardized turbulent wind characteristics under the analytical framework for typhoons and hurricanes were compared with the corresponding recommendations in standard of American Society of Civil Engineers (ASCE 7-10) and Architectural Institute of Japan Recommendations for Loads on Buildings (AIJ-RLB-2004).

**Keywords:** wind characteristics; boundary layer; typhoon; hurricane; field measurement

---

## 1. Introduction

Wind characteristics (e.g., mean wind speed, gust factor, turbulence intensity and integral scale, etc.) are the critical factors for wind-resistant design of the wind-sensitive infrastructures and urban planning. Resisting wind effects and reducing wind-induced damage in tropical cyclones is the challenge for the wind sensitive buildings and structures in cyclone-prone regions, as these

regions are normally in the economically developed areas with crowded populations and large-scale landmark buildings and structures. Therefore, a rational analytical framework for investigating wind characteristics in tropical cyclones is essential to understand the nature of winds, calibrate codes of practice for wind-resistant design of the large-scale structures and enhance wind tunnel simulations and numerical modeling [1]. Oncoming winds of buildings and bridges are usually simplified as the steady flow part featured by mean wind speed and corresponding vertical profile, and the fluctuating flow part characterized by turbulence intensity, integral scale, gust factor, peak factor, probability distribution, and power spectrum.

Tropical cyclones are characterized by the asymmetric helical flow structure and complex turbulence driven mechanism (both convective and mechanical turbulence). The spatial distribution of the flow structure also varies significantly in the footprint of tropical cyclones. Due to the limited measurements in the lower boundary layer of tropical cyclones, a basic premise of the existing codes and standards is that the turbulent wind characteristics in tropical cyclones are similar to those observed in the boundary layer winds of extratropical storms. However, it is well known that the downward transport of convective cells generated at higher levels together with the boundary layer rolls could modulate the wind structure and turbulence in the lower tropical cyclone boundary layer. These thermodynamics-related activities may lead to the turbulent wind characteristics of the hurricanes/typhoons different from those of the extratropical winds [2–4].

A direct and reliable approach to examine the turbulent wind characteristics is based on the field observations in the paths of landfalling tropical cyclones. Thus, a number of field measurement programs were initiated in the tropical cyclone-prone regions to monitor the hurricane/typhoon winds [5–14]. The field-measured wind characteristics from different observation stations for various tropical cyclones are quite scattered and hard-to-reach unified conclusions to guide the wind-resistance design of buildings and structures in the cyclone-prone regions. The inconsistent wind characteristics of tropical cyclones may be attributed in part to the complexity of turbulence driven mechanisms, e.g., shear (namely roll and streak structures near the surface), convection, rotation, blocking and sheltering effects at the boundary layer, and also the interactive motions of multi-scale eddies in the flow fields of tropical cyclones [15,16]. On the other hand, the underlying surface and the employed schemes to obtain the turbulence parameters may also significantly influence on the variability of wind characteristics. Since the tracks of tropical cyclones are random, most of the field observations were conducted by installing anemometers and accelerometers on structures or observation towers, which were built in the regions frequently attacked by tropical cyclones. The measured turbulent wind characteristics from these observation stations are quite different from one another because of the underlying surrounding terrain conditions and the lack of well-established guidelines for an appropriate documentation of the near surface wind filed in tropical cyclones. In the China wind codes, wind characteristics are specified over standard terrain with roughness length of 0.03 m, averaging time of 10 min for mean wind and duration time of 3 s for gusty wind at 10 m height. Accordingly, it is essential to convert the turbulence characteristics obtained from various stations to the standard terrain and investigate the wind nature in a unified analytical framework. The "standardized" wind characteristics due to their universality could be useful in instructing the structural design in cyclone-prone regions.

This study first presented an analytical framework in which the mean wind speed, turbulence intensity, integral scale, gust factor, and peak factor measured at various terrains, heights and averaging times were properly standardized. Then, the typhoon and hurricane wind data analyzed here were briefly described. Finally, field-observed turbulent wind characteristics of four typhoons and three hurricanes were converted to the standard condition and comparatively investigated. The standardized results were also compared with the corresponding recommendations in ASCE7-10 [17] and AIJ-RLB-2004 [18]. The difference of the wind characteristics in hurricanes and typhoons were also highlighted.

## 2. Analytical Framework

In this section, an analytical framework will be proposed in which the turbulent characteristics measured in various terrain conditions, heights and averaging times could be converted to a standard station. The standardization of the wind characteristics is based on the assumption of the equilibrium boundary-layer theory [19]. The atmosphere stratification in the boundary layer of tropical cyclones is assumed to be neutrally stable, which implies that the turbulent structure within this region is driven mainly by the local surface roughness effects [20]. In the non-equilibrium boundary-layer, this analytical framework may need further investigations subjecting to specific terrain conditions.

### 2.1. Mean Wind Speed

To analyze the wind characteristics, an essential step is to convert the wind speeds measured at different station conditions (i.e., various exposures, heights, and averaging times) to the standard condition. The standardization of the mean wind speed in this study follows the three steps: (1) determine the exposure type of the observation station; (2) calculate the gradient wind speeds over the observation exposure; and (3) calculate the mean wind speed at the reference height (10 m high) over standard exposure (open flat terrain) by assuming the gradient wind speeds are equal at the gradient height over different exposures.

#### 2.1.1. Logarithmic Law Wind Profile

Normally, the observations by Global Position System (GPS) dropsonde and Doppler radar show that the variation of mean wind speed with height follows the logarithmic law in the lower part of tropical cyclone boundary layer [21–25]. Thus, the logarithmic law can be used to describe lower boundary layer and the outer-vortex regions of a tropical cyclone:

$$U_s(z_s) = \frac{u_{*s}}{k} \ln\left(\frac{z_s}{z_{0s}}\right), \tag{1}$$

where $U_s(z_s)$ represents the mean wind speed at height $z_s$ over the standard exposure. Specifically, the standard exposure in this study corresponds to the roughness length $z_{0s} = 0.03$ m, the reference height is 10 m, and the time scale for the average value is 10 min. $u_{*s}$ denotes the friction velocity over the standard exposure, and $k \approx 0.40$ is the von Kármán constant.

According to Equation (1), the key procedure to standardize the mean wind speed is to determine the relationship of the friction velocities in various terrains. As all anemometers used in this study are set between 10–60 m height, it is reasonable to assume that the friction velocity in the lower tropical cyclone boundary layer is a height-independent constant [25–27]. Based on the assumption of local equilibrium conditions, the transition model in Engineering Sciences Data Unit (ESDU) [28], which has been applied to convert the 3 s peak speed over open-terrain and the 1-min mean wind speed above open water in hurricane by Simiu et al. [29], is employed:

$$\frac{u_{*s}}{u_{*m}} = \frac{\ln\left(\frac{10^5}{z_{0s}}\right)}{\ln\left(\frac{10^5}{z_{0m}}\right)}, \tag{2}$$

where $u_{*m}$ is the friction velocity over the field measured exposure with roughness length of $z_{0m}$. Then the relation of mean wind speeds with different terrains can be accordingly expressed as:

$$\frac{U_s(z_s)}{U_m(z_m)} = \frac{\ln\left(\frac{10^5}{z_{0s}}\right) \ln\left(\frac{z_s}{z_{0s}}\right)}{\ln\left(\frac{10^5}{z_{0m}}\right) \ln\left(\frac{z_m}{z_{0m}}\right)}, \tag{3}$$

where $U_m(z_m)$ is the mean wind speed measured at experiment station with height $z_m$ and roughness $z_{0m}$. In this model, the gradient balance assumption, which has been demonstrated to be valid at a

sufficiently high altitude [30], is adopted. The super-gradient flows were observed in the boundary layer of some tropical cyclones, however, it is not systematic in tropical cyclones, especially in overland conditions [31,32]. The case of super-gradient flows in tropical cyclones will discussed in next section.

The relation among mean wind speeds of various averaging times can be expressed as [33]:

$$U_\tau(z) = U_{3600}(z)\left[1 + \frac{\beta^{0.5}c(\tau)}{2.5\ln\left(\frac{z}{z_0}\right)}\right], \tag{4}$$

where $\tau$ denotes the averaging time; $U_\tau(z)$ and $U_{3600}(z)$ are respectively $\tau$-s mean and 1-h mean wind speeds; $\beta$ represents the ratio of the fluctuating wind speed variance to the square of friction velocity; $c(\tau)$ is an averaging time-related parameter that determined by statistical characteristics of wind speed measurements.

2.1.2. Super-Gradient Wind Profile

The field measurements show the existence of super-gradient wind over ocean surface and the sea land transition regions in tropical cyclones, and the variation of mean wind with height following a logarithmic-quadratic profile [34]. Based on the field measurements in hurricanes over land and ocean surface, Snaiki and Wu [35] proposed a semi-empirical model to depict the mean wind profile. As the empirical model is convenient and accurate, it is adopted here to convert the mean wind speed in landfalling typhoons. The power law-based wind profile is used as follows:

$$U_s(z_s) = U_{10_s}\left[\left(\frac{z_s}{10}\right)^{\alpha_s} + \eta_1 \sin\left(\frac{z_s}{\delta_s}\right)\exp\left(-\frac{z_s}{\delta_s}\right)\right], \tag{5}$$

where $U_s(z_s)$ and $U_{10_s}$ are the mean wind speed at height $z_s$ and 10 m over the standard exposure; $\alpha_s$ is the power law exponent over the standard exposure; $\delta_s$ is the height of the wind maximum over the standard exposure; $\eta_{1s}$ is derived to be:

$$\eta_{1s} = \frac{\left(\frac{\delta_s}{10}\right)^{\alpha_s}\alpha_s e}{\sin 1 - \cos 1},$$

Analogously, the field measured mean wind over the experiment exposure is:

$$U_m(z_m) = U_{10_m}\left[\left(\frac{z_m}{10}\right)^{\alpha_m} + \eta_{1m} \sin\left(\frac{z_m}{\delta_m}\right)\exp\left(-\frac{z_m}{\delta_m}\right)\right], \tag{6}$$

where $U_m(z_m)$ and $U_{10_m}$ are the mean wind speed at height $z_m$ and 10 m over the measured exposure; $\alpha_m$ is the power law exponent over the measured exposure; $\delta_m$ is the height of the wind maximum over the measured exposure; $\eta_{1s}$ is:

$$\eta_{1m} = \frac{\left(\frac{\delta_m}{10}\right)^{\alpha_m}\alpha_m e}{\sin 1 - \cos 1},$$

By adopting the assumption that the wind speeds at the wind maximum height ($\delta_s$ and $\delta_m$) are equal, the following expression can be deduced:

$$\frac{U_s(z)}{U_m(z)} = \frac{\left(\frac{\delta_m}{10}\right)^{\alpha_m}\left[\left(\frac{z_s}{10}\right)^{\alpha_s} + \eta_{1s} \sin\left(\frac{z_s}{\delta_s}\right)\exp\left(-\frac{z_s}{\delta_s}\right)\right]}{\left(\frac{\delta_s}{10}\right)^{\alpha_s}\left[\left(\frac{z_m}{10}\right)^{\alpha_m} + \eta_{1m} \sin\left(\frac{z_m}{\delta_m}\right)\exp\left(-\frac{z_m}{\delta_m}\right)\right]}, \tag{7}$$

Equation (7) could be used to convert the field wind speeds to the standard exposures in the tropical cyclones with super-gradient flow. Actually when the wind speed measured in the lower regions following the logarithmic law, the Equation (7) will merge into the logarithmic law or power law.

## 2.2. Turbulence Intensity

It is conventional to treat the turbulence ratio (the ratio of the standard deviation of longitudinal wind velocity component $\sigma_u$ to the friction velocity $u_*$) as terrain-independent in the equilibrium boundary layer [36,37]. On the other hand, Harris and Deaves [38] proposed an empirical model to consider the variation of turbulence ratio with height as:

$$\frac{\sigma_u}{u_*} = 2.63\eta\left[0.538 + 0.090\ln\left(\frac{z}{z_0}\right)\right]^{\eta^{16}},\tag{8}$$

where $\eta = 1 - z/h$; $h = u_*/(6f)$; $f = 1.458 \times 10^{-4}\sin\phi$ is the Coriolis parameter; and $\phi$ denotes the latitude of the observation site. Due to the fact that the derivation of Equation (8) was partly based on non-equilibrium-condition data, the estimation of maximum turbulence ratio, $[\sigma_u/u_*]_{max}$ obtained from Equation (8) is dependent on the terrain roughness length, which is in contradiction to the equilibrium assumption. To correct this issue, the empirical variation of $\sigma_u/u_*$ with respect to terrain roughness is introduced in ESDU [39] to obtain approximately a constant $[\sigma_u/u_*]_{max}$ for various terrain roughness lengths:

$$\left[\frac{\sigma_u}{u_*}\right](z_0) = 1 + 0.156\ln\left(\frac{u_*}{fz_0}\right),\tag{9}$$

Since the field measurements give $[\sigma_u/u_*]_{max} = 2.85$ for the terrain with a roughness length of 0.03 m, Equation (8) can be corrected by factoring $2.85/\{1 + 0.156\ln[u_*/(fz_0)]\}$, resulting in an improved model to calculate turbulence ratio as in ESDU [39]:

$$\frac{\sigma_u}{u_*} = \frac{7.496\eta\left[0.538 + 0.090\ln\left(\frac{z}{z_0}\right)\right]^{\eta^{16}}}{1 + 0.156\ln\left(\frac{u_*}{fz_0}\right)}.\tag{10}$$

As expected, Equation (9) gives a maximum turbulence ratio $[\sigma_u/u_*]_{max}$ of approximately 2.85 for various roughness lengths. However, field measurements show that turbulence ratios in tropical cyclones are usually greater than the values in extratropical storms [12,24,40], making the selection of $[\sigma_u/u_*]_{max} = 2.85$ inapplicable to tropical cyclones. On the other hand, a height-independent relation between turbulence ratio $\sigma_u/u_*$ and underlying surface roughness length $z_0$ was proposed by Li et al. [13] based on the analysis of field measurements in typhoons. In this study, this height-independent relation proposed by Li et al. [13] is adopted as:

$$\left[\frac{\sigma_u}{u_*}\right]_{max} = 2.72 - 0.25\log z_0.\tag{11}$$

As a result, the turbulence ratio in tropical cyclone will be corrected by multiplying Equation (8) by the following factor:

$$\frac{2.72 - 0.25\log z_0}{1 + 0.156\ln[u_*/(fz_0)]}.\tag{12}$$

Then the turbulence ratio could be expressed as:

$$\frac{\sigma_u}{u_*} = \frac{2.63\eta\left[0.538 + 0.009\ln\left(\frac{z}{z_0}\right)\right]^{\eta^{16}}[2.72 - 0.25\log(z_0)]}{1 + 0.156\ln\left(\frac{u_*}{fz_0}\right)}.\tag{13}$$

For a standard terrain condition ($z = 10$ m; $z_0 = 0.03$ m; assuming $u_* = 1$ m/s and $\phi = 25°$), the turbulence ratios estimated by Equations (9) and (11) are 2.55 and 2.78, respectively. In this

study, Equation (13) is utilized to convert the measured turbulence ratio to the standard condition. Accordingly, the longitudinal turbulence intensity in a standard exposure can be calculated as:

$$[I_u]_s = \frac{2.63\eta_s\left[0.538 + 0.090\ln\left(\frac{z}{z_0}\right)\right]^{\eta_s^{16}}[2.72 - 0.25\ln(z_{0s})][u_*]_s}{\left[1 + 0.156\ln\left(\frac{u_*}{fz_0}\right)\right][U]_s}.$$

(14)

### 2.3. Integral Scale

The approach of integrating correlation function by invoking Taylor's hypothesis is frequently used to estimate the integral scale as it has a clear physical meaning [24,33]:

$$L_u^x = \frac{U}{\sigma_u^2}\int_0^{R_{uu}=0.05\sigma_u} R_{uu}(\tau)d\tau,$$

(15)

where $R_{uu}$ is the autocorrelation function of the longitudinal fluctuating component.

The integration of autocorrelation function, however, usually overestimates the value of integral scale and will result in a deviation of inertial sub-range in the estimated von Kármán-type spectrum [41] compared to that in the field-measured spectra. To improve the accuracy, Harris and Deaves [38] suggested the following model to estimate the longitudinal integral scale:

$$L_u^x = \frac{A^{\frac{3}{2}}\left(\frac{\sigma_u}{u_*}\right)^3 z}{2.5K_z^{\frac{3}{2}}\left(1 - \frac{z}{h}\right)^2\left(1 + 5.75\frac{z}{h}\right)},$$

(16)

where $z$ is the height from the ground.

$$A = 0.115\left(1 + 0.315\eta^6\right)^{\frac{2}{3}}$$

(17)

and

$$K_z = 0.19 - (0.19 - K_0)\exp\left[-B\left(\frac{z}{h}\right)^N\right]$$

(18)

in which

$$K_0 = \frac{0.39}{R_0^{0.11}},$$

(19)

$$B = 24R_0^{0.155},$$

(20)

$$N = 1.24R_0^{0.008},$$

(21)

$$R_o = \frac{u_*}{fz_0}.$$

(22)

The longitudinal integral scale over standard exposure $[L_u^x]_s$ can be estimated according to Equation (16) by introducing the corresponding values of $[u_*]_s$ and $[z_0]_s$.

### 2.4. Peak Factor

Peak factor $g_u$ is defined as the ratio of maximum wind speed fluctuation in a duration $\tau$ to the standard deviation of the fluctuating wind speed within an observation period of $T$:

$$g_u(\tau, T) = \frac{\max[u(\tau, T)]}{\sigma_u(\tau, T)}\frac{\sigma_u(\tau, T)}{\sigma_u}.$$

(23)

For a stationary stochastic process following Gaussian distribution, the peak factor with $\tau \to 0$ and $T \geq 3600\ s$ could be calculated as [42]:

$$g_u(\tau, T) = \sqrt{2\ln[\nu(\tau, T)T]} + \frac{0.5772}{\sqrt{2\ln[\nu(\tau, T)T]}}, \tag{24}$$

where $\nu(\tau, T)$ is the zero up-crossing rate. It can be calculated by [42,43]:

$$\nu^2(\tau, T) = \frac{\int_0^\infty n^2 S_u(n) \chi^2(n, \tau, T) \mathrm{d}n}{\int_0^\infty S_u(n) \chi^2(n, \tau, T) \mathrm{d}n}, \tag{25}$$

in which $S_u(n)$ represents the wind velocity spectrum; $n$ denotes the frequency in Hertz, and $\chi^2(n, \tau, T)$ is a filter function used to consider the influence of sampling frequency, averaging time and response characteristics of the anemometer. In this study, the von Kármán-type spectrum is employed as:

$$\frac{nS_u(n)}{\sigma_u^2} = \frac{4\left(\frac{nL_u^x}{U}\right)}{\left[1 + 70.8\left(\frac{nL_u^x}{U}\right)^2\right]^{\frac{5}{6}}}, \tag{26}$$

The filter function is chosen as following for sonic anemometers [44]:

$$\chi^2(n, \tau, T) = \left[\frac{\sin(\pi n \tau)}{\pi n \tau}\right]^2 - \left[\frac{\sin(\pi n T)}{\pi n T}\right]^2, \tag{27}$$

For propeller anemometers, the following filter function, which takes the mechanical features of propeller anemometers into consideration, is adopted [43]:

$$\chi^2(n, \tau, T) = \left\{\left[\frac{\sin(\pi n \tau)}{\pi n \tau}\right]^2 - \left[\frac{\sin(\pi n T)}{\pi n T}\right]^2\right\} \frac{1}{1 + \left(\frac{2\pi n \lambda}{U}\right)^2}, \tag{28}$$

where $\lambda$ is the distance constant of the propeller anemometer.

Equation (24) is valid for calculating the average of instantaneous peak factor ($\tau \to 0$) from a long enough wind speed record (e.g., $T \geq 1$ h). With a finite averaging time, $\tau$, and a finite observation period, $T$, the estimation of standard deviation in Equation (26) could be biased since the measured spectrum is truncated in both high-frequency and low-frequency regions, and might eventually lead to an inaccurate estimation of the peak factor. In the case that these conditions are not satisfied, the following relation is necessary to be introduced to consider the effects of the variance reduction due to the truncation of the velocity spectrum:

$$\frac{\sigma_u(\tau, T)}{\sigma_u} = \frac{\int_0^\infty S_u(n) \chi^2(n, \tau, T) \mathrm{d}n}{\int_0^\infty S_u(n) \mathrm{d}n}, \tag{29}$$

The 3-s peak factor, $[g_u]_s$, in time scale $[T]_s = 600$ s in the standard terrain can be estimated by introducing $U_s$ and $L_{us}^x$, which were respectively calculated through Equations (3) or (7) and (16).

*2.5. Gust Factor*

Gust factor, $G_u(\tau, T)$, herein is defined as the ratio of gust speed with gust duration $\tau$ to the mean wind speed $U(T)$ with an observation period of $T$:

$$G_u(\tau, T) = 1 + \frac{\max[u(\tau, T)]}{\sigma_u(\tau, T)} \frac{\sigma_u(\tau, T)}{\sigma_u(\Delta t, T)} \frac{\sigma_u(\Delta t, T)}{U(T)}, \tag{30}$$

where $\Delta t$ is the sampling interval.

Substituting the peak factor and turbulence intensity into the corresponding terms of Equation (30), the gust factor can be re-expressed as:

$$G_u(\tau, T) = 1 + g_u(\tau, T)I_u \frac{\sigma_u(\tau, T)}{\sigma_u(\Delta t, T)},\tag{31}$$

where $\sigma_u(\tau, T)/\sigma_u(\Delta t, T)$ can be calculated by:

$$\frac{\sigma_u(\tau, T)}{\sigma_u(\Delta t, T)} = \frac{\int_0^\infty S_u(n)\chi^2(n, \tau, T)\mathrm{d}n}{\int_0^\infty S_u(n)\chi^2(n, \Delta t, T)\mathrm{d}n},\tag{32}$$

As a result, the gust factor in the standard exposure, $[G_u(\tau, T)]_s$, can be estimated by:

$$[G_u(\tau, T)]_s = 1 + [g_u(\tau, T)]_s[I_u]_s\left[\frac{\sigma_u(\tau, T)}{\sigma_u(\Delta t, T)}\right]_s,\tag{33}$$

## 3. Data Sources

### 3.1. Tropical Cyclones and Instruments

In this study, the data of four typhoons (0601 typhoon Chanchu, 0606 typhoon Prapiroon, 0812 typhoon Nuri, and 0814 typhoon Hagupit) and three hurricanes (0504 hurricane Katrina, 0510 hurricane Rita, and 0512 hurricane Wilma) were comparatively analyzed. The detailed descriptions of the observation site exposures and the observation tower configurations for the four typhoons and three hurricanes were presented in Li et al. [16] and Masters et al. [12], respectively. The GPS coordinates of the observation stations were listed in Table 1. As the latitudes of all observation stations are around 25°, the latitude of the standard condition is set to be 25° for the convenient of calculation.

**Table 1.** The GPS coordinates of observation towers. Reproduced with permission from [16], Elsevier, 2019.

| Tropical Cyclones | Tower | Latitude | Longitude |
| --- | --- | --- | --- |
| Chanchu | RBT | 22.7337° | 115.5734° |
| | OT | 23.5510° | 117.0020° |
| Prapiroon | BT | 21.4519° | 111.3149° |
| Nuri | MFB | 22.1810° | 113.5630° |
| | DIT | 22.1413° | 113.7096° |
| Hagupit | ZT | 21.4509° | 111.3745° |
| | ST | 21.2538° | 110.6541° |
| Katrina | T1 | 29.8253° | −90.0319° |
| | T2 | 29.4441° | −90.2628° |
| | T3 | 30.4720° | −88.5308° |
| Rita | T0 | 29.9512° | −94.0220° |
| | T3 | 29.9548° | −93.9542° |
| | T5 | 30.0797° | −93.7841° |
| Wilma | T0 | 25.9008° | −81.3114° |
| | T1 | 26.1458° | −80.5067° |
| | T2 | 25.8681° | −80.8997° |
| | T3 | 25.7516° | −80.3780° |

It is noted that the distance constant $\lambda$ of the propeller anemometer is an important factor to calculate the peak factor and gust factor as this type of anemometer mechanically filters the amplitudes

of gusts with wavelengths less than $2\pi\lambda$ due to the mechanical limitations [10]. In this study, the propeller anemometers of models R.M. Young 05103L and R.M. Young 27106R were respectively used to measure typhoons and hurricanes. The specifications of these two propeller anemometers are listed in Table 2. Based on the parameters in Table 2, the data measured by propeller anemometers were corrected according to Equation (24). In addition to propeller anemometers, the sonic anemometers were also utilized in the field measurement of typhoons. Specifically, two 3-D ultrasonic anemometers (WindMaster™ Pro., Gill Instruments Ltd., Lymington, UK) were installed on tower RBT and one 3-D ultrasonic anemometer (HD2003, Delta Ohm Srl, Selvazzano Dentro, Italy) were setup on tower OT for the measurements of Typhoon Chanchu; one HD2003 anemometer were installed on tower BT to acquire data from Typhoon Prapiroon, and towers DIT and ST were equipped with Gill WindMaster™ Pro. anemometers. The specifications of the utilized sonic anemometers are also listed in Table 2. For the data obtained from sonic anemometers, the filter function presented by Equation (23) were used for the correction.

**Table 2.** Specifications of anemometers.

| Anemometers | Specifications | | |
|---|---|---|---|
| 05103L | Wind speed | Range | 0~100 m/s |
| | | Threshold Sensitivity | 1 m/s |
| | | Distance constant | 2.7 m for 63% recovery |
| | Wind direction | Ranges | 0 ~ 360° |
| | | Threshold Sensitivity | 1.1 m/s at 10° displacement |
| | | Delay Distance | 1.3 m for 50% recovery |
| | | Damped Natural Wavelength | 7.4 m |
| 27106R | Wind speed | Range | 0~25 m/s |
| | | Threshold Sensitivity | 0.3 m/s |
| | | Distance constant | 2.7 m for 63% recovery |
| | | Damped Natural Wavelength | 7.4 m |
| WindMaster™ Pro | Wind speed | Range | 0~ 65 m/s |
| | | Resolution | 0.01 m/s |
| | Wind direction | Ranges | 0~359° |
| | | Resolution | 0.1° |
| HD2003 | Wind speed | Range | 0~60 m/s |
| | | Resolution | 0.01 m/s |
| | Wind direction | Ranges | 0~359° |
| | | Resolution | 0.1° |

### 3.2. Data Quality Control and Data Source

Tropical cyclones are characterized by strong winds accompanied by torrential rain, ocean waves, and storm surge. The representative wind records are usually located in the eyewall regions of tropical cyclones. In the field measurements of tropical cyclones, however, the anemometers in heavy rain bands usually present some spikes and errors. Therefore, the data quality-control procedure is a necessary step before the analysis of the wind characteristics. In this study, the data quality-control schemes and the criteria for the selection of samples were referred to Li et al. [13]. Specifically, the spikes and errors in the data were first identified and replaced by the five-point weighted averages. Then, the reverse arrangement test [45] and run test [46] with a 95% significance level were employed to test the stationarity of the recorded winds. The datasets that failed to pass both two types of stationary tests were removed from the analysis. The stationarity test ensured that the analyzed data could satisfy the local equilibrium boundary-layer assumption, where the friction velocity is independent of the location in the along-wind direction and the Reynolds number [37].

Table 3 briefly summarized the datasets utilized in this study, together with the observation heights, types of the anemometers used, number of the runs in each group (sample size) and average of the 10 min mean wind speeds. The detailed analysis of the turbulent wind characteristics of the original datasets both in typhoons and hurricanes were presented in Li et al. [16].

**Table 3.** Datasets analyzed in this study. Reproduced with permission from [16], Elsevier, 2019.

| Tropical Cyclones | Sites | Height (m) | Anemometer Types | Number of Runs | Average of the 10 min Mean Wind Speed (m/s) |
|---|---|---|---|---|---|
| Chanchu | RBT | 10 | Sonic | 26 | 19.19 |
| | | | Propeller | 15 | 18.87 |
| | | 30 | Sonic | 37 | 21.96 |
| | | 60 | Propeller | 58 | 22.19 |
| | OT | 5 | Sonic | 21 | 24.69 |
| | | 10 | Sonic | 45 | 24.28 |
| Prapiroon | BT | 10 | Sonic | 18 | 20.13 |
| | | | Propeller | 4 | 22.69 |
| Nuri | MFB | 30 | Sonic | 8 | 18.25 |
| | DIT | 10 | Sonic | 83 | 24.26 |
| | | 60 | Sonic | 100 | 25.11 |
| Hagupit | ZT | 60 | Sonic | 38 | 28.53 |
| | ST | 5 | Sonic | 18 | 20.34 |
| | | | Propeller | 27 | 20.34 |
| | | 10 | Sonic | 61 | 21.30 |
| | | | Propeller | 58 | 20.93 |
| Katrina | T1 | 5 | Propeller | 16 | 21.98 |
| | | 10 | Propeller | 43 | 22.45 |
| | T2 | 5 | Propeller | 34 | 21.50 |
| | | 10 | Propeller | 40 | 24.42 |
| | T3 | 5 | Propeller | 5 | 23.02 |
| | | 10 | Propeller | 14 | 22.46 |
| Rita | T0 | 5 | Propeller | 46 | 21.05 |
| | | 10 | Propeller | 60 | 22.17 |
| | T3 | 10 | Propeller | 15 | 18.85 |
| | T5 | 5 | Propeller | 19 | 18.86 |
| | | 10 | Propeller | 32 | 19.55 |
| Wilma | T0 | 5 | Propeller | 13 | 21.08 |
| | | 10 | Propeller | 19 | 22.41 |
| | T1 | 5 | Propeller | 23 | 23.65 |
| | | 10 | Propeller | 37 | 26.17 |
| | T2 | 5 | Propeller | 16 | 21.34 |
| | | 10 | Propeller | 18 | 23.66 |
| | T3 | 5 | Propeller | 9 | 21.36 |
| | | 10 | Propeller | 14 | 22.63 |

## 4. Results and Discussions

The selected datasets from the four typhoons and the three hurricanes summarized in the preceding section were investigated in the analytical framework presented in Section 2. Specifically, both the datasets in typhoons and hurricanes were converted to the standard exposure with a roughness length

of 0.03 m at 10 m height and an observation time scale of 10 min. The latitude of the standard terrain is assumed to be 25°.

## 4.1. Turbulence Intensity

The turbulence intensities were extracted based on the analytical framework and shown in Figures 1a and 2a respectively for investigated typhoons and hurricanes. The corresponding probability density functions (PDFs) are shown in Figures 1b and 2b. For typhoons, the average value of longitudinal turbulence intensities is 0.1952 and the standard deviation is 0.0032. For hurricanes, the average value of longitudinal turbulence intensities is 0.1906 and the standard deviation is 0.0022. The longitudinal turbulence intensity of these four typhoons presents slightly higher values in terms of both mean and standard deviation compared to those three hurricanes. This observation can be further demonstrated by comparing Figures 1b and 2b, where the probability distribution of the longitudinal turbulence intensities in both typhoons and hurricanes follows the normal distribution quite well. One can easily conclude that in these four typhoons the turbulence intensity has a larger value than that in those three hurricanes under the same probability of exceedance.

In ASCE7-10, the longitudinal turbulence intensity is given by:

$$I_u = c\left(\frac{10}{z}\right)^{\frac{1}{6}}, \tag{34}$$

where $c$ equals to 0.30, 0.20, and 0.15 for category B, C (corresponding to the standard terrain in this study), and D exposures, respectively. In AIJ-RLB-2004 code, the longitudinal turbulence intensity over flat terrain categories is given by:

$$I_u = 0.1\left(\frac{z}{z_G}\right)^{-\alpha-0.05}, \tag{35}$$

in which $\alpha$ and $z_G$ are parameters reflecting the category of exposures. In category II exposure, which is the closest to the standard terrain in this study, $\alpha$ and $z_G$ are respectively 0.15 and 350. The longitudinal turbulence intensities obtained from ASCE7-10 and AIJ-RLB-2004 are 0.2000 and 0.2036, respectively, for the standard exposure. As depicted in Figures 1a and 2a, both ASCE7-10 and AIJ-RLB-2004 present a slightly higher estimation of longitudinal turbulence intensity for hurricanes and typhoons. Generally, the estimation of ASCE7-10 is relatively better compared to that of AIJ-RLB-2004.

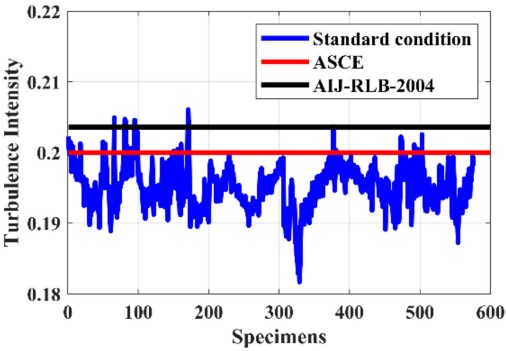

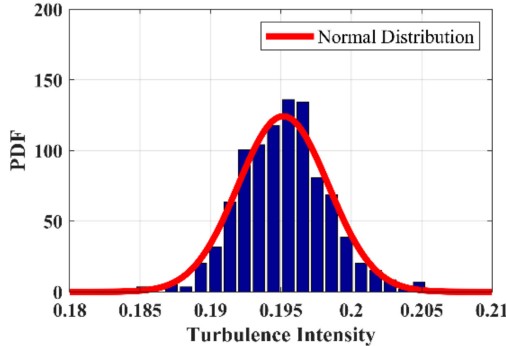

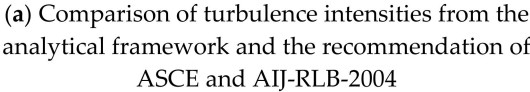

(**a**) Comparison of turbulence intensities from the analytical framework and the recommendation of ASCE and AIJ-RLB-2004

(**b**) PDF of the turbulence intensities from the analytical framework

**Figure 1.** Turbulence intensities and their probability density functions (PDF) for typhoons.

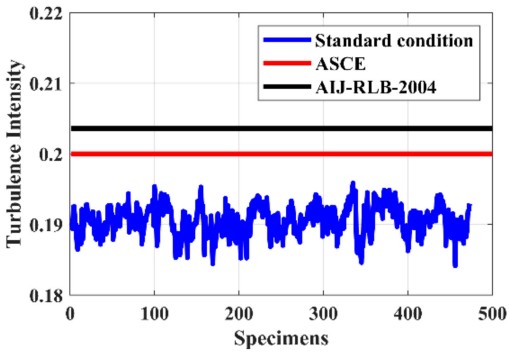

(**a**) Comparison of turbulence intensities from the analytical framework and the recommendation of ASCE and AIJ-RLB-2004

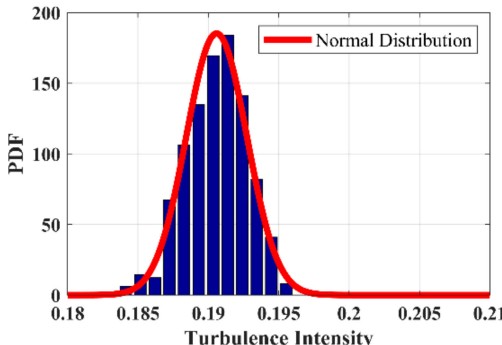

(**b**) PDF of the turbulence intensities from the analytical framework

**Figure 2.** Turbulence intensities and their PDF for hurricanes.

### 4.2. Integral Scale

The integral scales and the corresponding probability distributions obtained are presented in Figures 3 and 4 for typhoons and hurricanes, respectively. The longitudinal integral scale in typhoons has an average value of 146.4482 and a standard deviation of 9.2143. The length scales extracted from hurricane measurements have a slightly higher average value of 157.5796 and a significantly lower standard deviation of 3.8106 compared to those of typhoons. As shown in Figures 3a and 4a, the range of integral scales extracted from typhoon measurements is significantly larger than that of hurricanes, which can be better illustrated by the probability distributions of longitudinal integral scales in typhoons and hurricanes. The probability distribution of typhoons follows the Weibull distribution with scale parameter of 150.368 and shape parameter of 20.0454, while the probability distribution of longitudinal integral scales for hurricanes follows the generalized extreme value distribution with scale parameter of −0.3259, shape parameter of 3.8960 and location factor of 156.311. Compared with Figure 4b, where the observed hurricane length scale shows a narrow distribution, the probability distribution of the observed typhoon integral scale of Figure 3b has a significantly wider range. This phenomenon may be in part attributed to the exposures of the observation station for the original datasets. The observation station in those three hurricanes are located in homogeneous open flat terrain as stated in Masters et al. [12]. However, in the observation of those four typhoons, the exposures of the measured stations are a little bit inhomogeneous. Another influence could be ascribed to the differences of turbulent structures of those typhoons and hurricanes. As noted in Li et al. [16], at the same roughness regime, the field measured integral scales in these four typhoons were greater than that in those three hurricanes. The different distributions of the observed hurricane and typhoon length scales might indicate that energy-containing eddies in the observed typhoons have various representative length-scales while those of the observed hurricanes are concentrated around the mean value. The multiple-scale eddy interactions in typhoons and hurricanes need further investigations before fully understanding the observed difference.

In ASCE7-10, the longitudinal integral scale is computed by:

$$L_u^x = l\left(\frac{z}{10}\right)^{\bar{\varepsilon}},\tag{36}$$

where $l$ and $\bar{\varepsilon}$ are respectively 152.4 m and 0.2 for category C exposure (standard exposure in this study). In AIJ-RLB-2004 code, the turbulence integral scale is defined independently of the terrain categories and is given by:

$$L_u^x = \begin{cases} 100\left(\frac{z}{30}\right)^{0.5} & 30\,\text{m} < z < z_G \\ 100 & z \leq 30\,\text{m} \end{cases},\tag{37}$$

where $z_G$ equals to 350 m for category II exposure, corresponding to the standard exposure in this study. Accordingly, the longitudinal integral scales obtained from ASCE7-10 and AIJ-RLB-2004 are respectively, 152.4 m and 100 m, for the standard exposure. It is noted that ASCE7-10 presents a reasonable estimation of the longitudinal integral scales for both typhoons and hurricanes. However, AIJ-RLB-2004 underestimates the longitudinal integral scales for both typhoons and hurricanes, suggesting that the usage of AIJ-RLB-2004 may lead to an inaccurate estimation of the power spectrum.

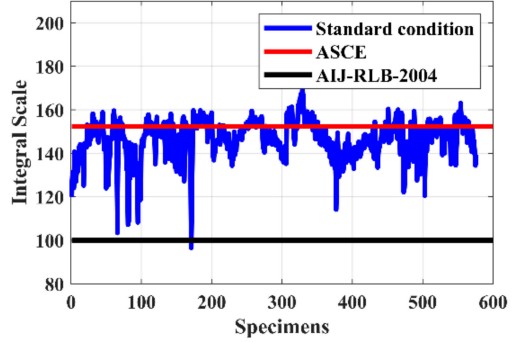

(**a**) Comparison of longitudinal Integral scales from the analytical framework and the recommendation of ASCE and AIJ-RLB-2004

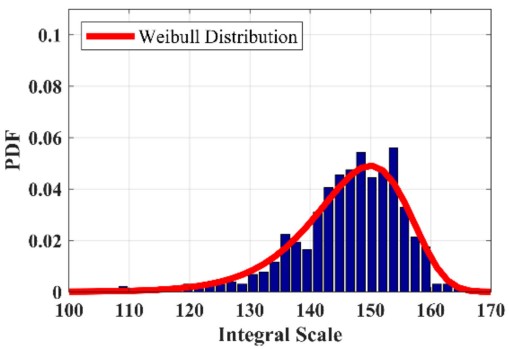

(**b**) PDF of the longitudinal integral scales

**Figure 3.** Longitudinal integral scales and their PDF for typhoons.

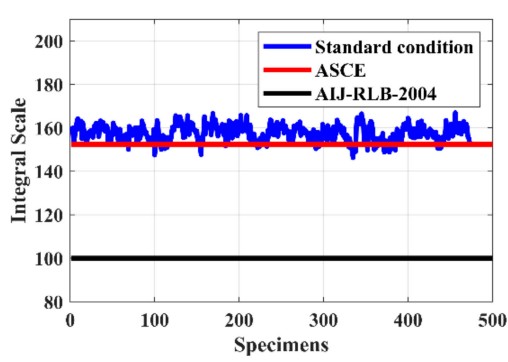

(**a**) Comparison of longitudinal Integral scales from the analytical framework and the recommendation of ASCE and AIJ-RLB-2004

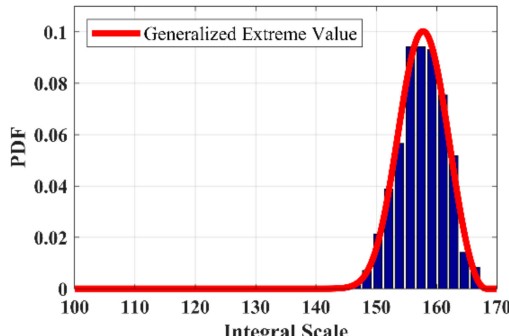

(**b**) PDF of the longitudinal integral scales

**Figure 4.** Longitudinal integral scales and their PDF for hurricanes.

### 4.3. Peak Factor

The peak factor is usually utilized for the estimation of gust factor, which plays an important role in determining the wind load on structures [43]. The estimated peak factors of typhoons and hurricanes are respectively presented in Figures 5 and 6, together with the corresponding probability distributions. For typhoons, the average value of the peak factor is 2.5211 and the standard deviation is 0.0198. The fitted PDF is shown in Figure 5b. For hurricanes, the average value of peak factor is 2.5123, slightly smaller than that of typhoons, and the standard deviation of peak factor is 0.0102, significantly smaller than that of typhoons. The probability distribution of peak factors in hurricanes follows t location-scale distribution with location parameter of 2.5174, scale parameter of 0.0022 and shape parameter of 0.9845.

Neither the expression nor the value of peak factor is explicitly prescribed in ASCE7-10. By matching the gust factor over open terrain ($G_u = 1.53$) and the turbulence intensity $I_u$ of

0.2 in Equation (34), a peak factor of 2.65 could be obtained. It should be noted that this calculation is based on an averaging time of 1 h. For a duration of $t_g$ , the gust wind speed can be expressed as:

$$\hat{U}(t_g, T) = \overline{U}(T) + g_u(t_g, T)\sigma_u. \tag{38}$$

Suppose the turbulent wind fluctuations follow the Gaussian distribution, the peak factor will be associated with the exceedance probability of the standard normal distribution. The probability of exceedance of wind gust with a duration of $t_g$ within an observation period of $T$ could be calculated as [47,48]:

$$P[U > \hat{U}(t_g, T)] = \frac{t_g}{T}. \tag{39}$$

Thus, the peak factor should satisfy:

$$g_u(t_g, T) = \Phi^{-1}\left(1 - \frac{t_g}{T}\right). \tag{40}$$

With the gust duration of 3 s and the averaging time of 10 min, the peak factor is around 2.575, which is slightly higher than the measured values in typhoons and hurricanes. AIJ-RLB-2004 carries out a performance-based wind resistant design procedure. Accordingly, the peak factor is included in the required performance of wind load level and the return period of wind speed. Hence, the comparison of the measurement results with AIJ-RLB-2004 is not discussed here.

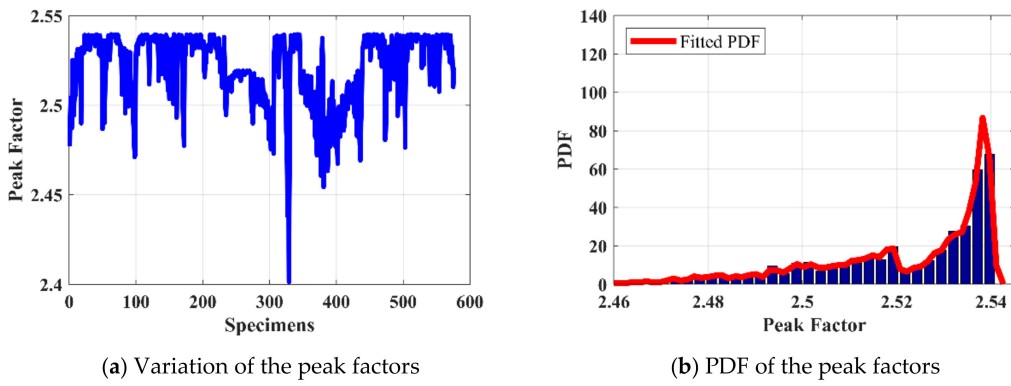

(**a**) Variation of the peak factors (**b**) PDF of the peak factors

**Figure 5.** Peak factors and their PDF for typhoons.

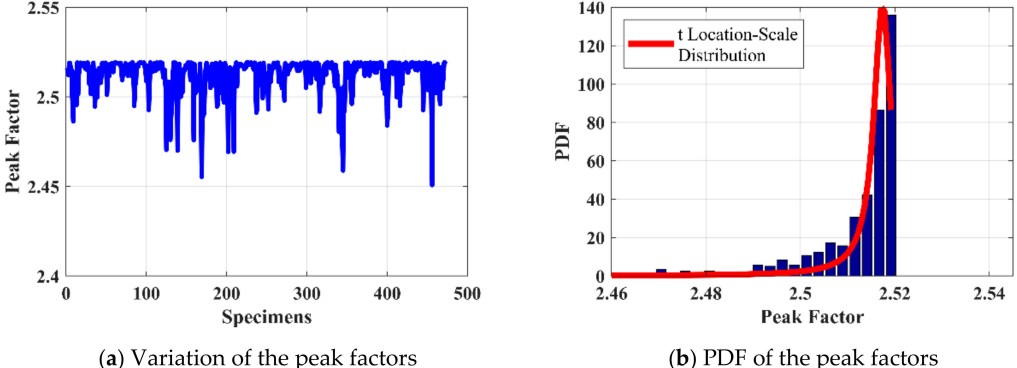

(**a**) Variation of the peak factors (**b**) PDF of the peak factors

**Figure 6.** Peak factors and their PDF for hurricanes.

*4.4. Gust Factor*

The gust factor in steady wind conditions depends on several wind characteristics, such as the intensity and integral scale, hence, it is a basic representation of the dynamic properties of wind loads [49]. Figure 7 depicts the gust factors and the corresponding probability distribution of typhoon

winds obtained based on the unified analysis framework. For typhoons, the average value of gust factors is 1.4919 and the standard deviation is 0.0069. The gust factors of hurricanes are presented in Figure 8a, where the gust factors have a mean of 1.4787 and a standard deviation of 0.0071. The mean of gust factors for typhoons are slightly higher than that for hurricanes, while the standard deviations of gust factors for typhoons and hurricanes are almost identical with similar probability distribution shapes. The probability distribution of gust factors for typhoons follows the extreme value distribution with a location parameter of 1.4948 and a scale parameter of 0.0051, while the probability distribution of gust factors for hurricanes follows the generalized extreme value distribution with shape parameter of $-0.5535$, location parameter of 1.4771, and scale parameter of 0.0076.

In ASCE7-10, the calculation of gust factor is referenced to the gust factor curve proposed by Durst [50]. The averaging time of the mean wind speed is 1 h in the Durst gust factor curve, while the gust factor is calculated based on an averaging time of 10 min in this study. Therefore, the conversion scheme for the gust factors with different averaging times presented in Vickery and Skerlj [20] was utilized here. Gust factor with a duration of 3 s and an averaging time of 10 min can be expressed as:

$$G_u = 1 + (SU)[SD(600, 3)], \tag{41}$$

where $SU$ is the value of the standard normal deviation associated with the exceedance probability of 0.5% and equals to 2.575. The $SD(600, 3)$ could be estimated by the following formula:

$$SD(600, 3) = \left[SD^2(3600, 3) - SD^2(3600, 600)\right]^{1/2}, \tag{42}$$

where $SD(3600, 3)$ and $SD(3600, 600)$ can be interpolated as indicated in Vickery and Skerlj [20] and equal to 0.1617 and 0.0650, respectively. The gust factor with duration of 3 s and averaging time of 10 min based on Equations (37) and (38) is around 1.3814, which indicates that the gust factors for both typhoons and hurricanes are greater than those for extratropical storms.

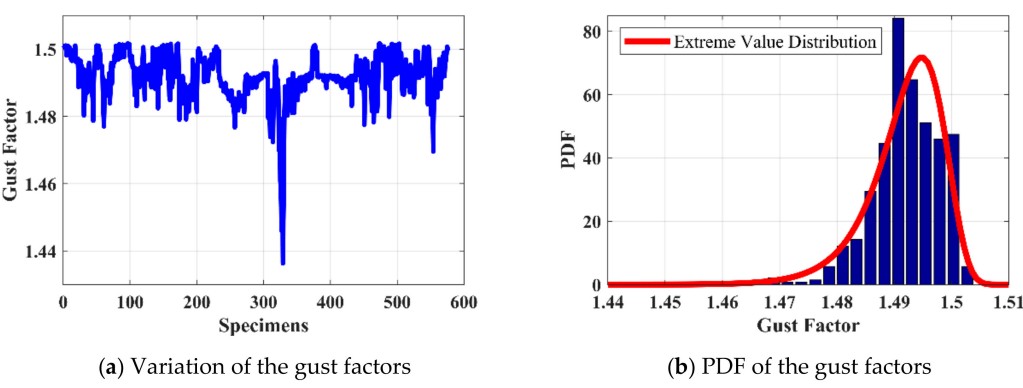

(**a**) Variation of the gust factors　　　　　　　　(**b**) PDF of the gust factors

**Figure 7.** Gust factors for typhoons.

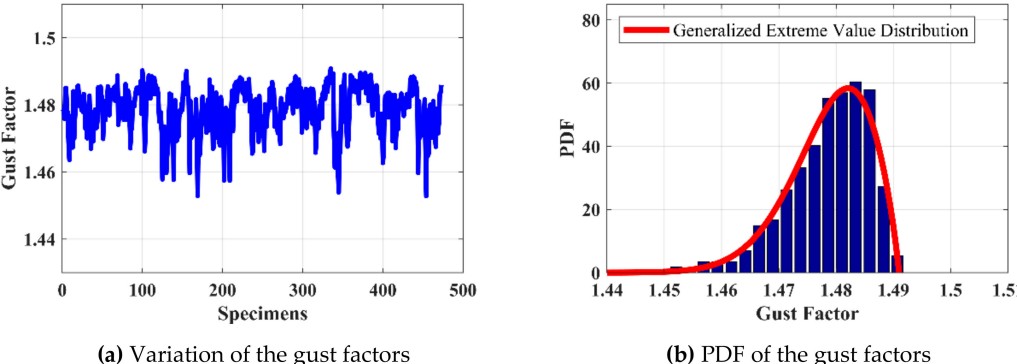

(**a**) Variation of the gust factors　　　　　　　　(**b**) PDF of the gust factors

**Figure 8.** Gust factors for hurricanes.

## 5. Concluding Remarks

An analytical framework was introduced in this study to standardize the turbulent wind characteristics, namely turbulence intensity, integral scale, peak factor and gust factor for various terrain conditions, heights, and averaging times in tropical cyclones. This analytical framework is based on the equilibrium boundary-layer theory and the assumption that the lower tropical cyclone boundary layer is neutrally stable. Field-measured data of the four typhoons and three hurricanes were standardized to the reference exposure with roughness length of 0.03 m, height of 10 m, and averaging time of 10 min, and then utilized for the extraction of wind characteristics under standard exposure. The differences of obtained wind characteristics between typhoons and hurricanes were highlighted, which may be attributed to the basins and latitudes of the genesis of hurricanes and typhoons, the influence of local topography and sea-land transition zone and the differences in turbulent flow structures of typhoons and hurricanes that need further investigations. More specifically, the wind characteristics of these observed typhoons typically present larger values compared to those of observed hurricanes, except for the turbulence integral scale. The turbulence integral lengths of typhoons have a wider distribution compared with those of hurricanes. The obtained turbulent wind characteristics based on the unified analysis framework were comparatively investigated together with the recommendations in ASCE7-10 and AIJ-RLB-2004. The difference between the standardized turbulent characteristics and the corresponding suggested values in the standards (ASCE7-10 and AIJ-RLB-2004) indicates that the tropical cyclone-induced wind loads need be taken into consideration in standards for tropical cyclone-prone regions. It is noted that the ASCE7-10 presents good estimations of the longitudinal turbulence intensity and integral scale for both typhoons and hurricanes, while the peak factor was slightly overestimated and the gust factor was underestimated. The AIJ-RLB-2004 makes a slightly higher estimation of the longitudinal turbulence intensity and a lower estimation of the longitudinal integral scale for both typhoons and hurricanes. The potential reason may be ascribed to the limitation of datasets which used to specify the wind characteristics, although it includes both tropical and extratropical winds. As noted in the AIJ-RLB-2004, the integral scale was treated to be terrain independent. However, the scales of wind eddies are strongly affected by the local roughness.

**Author Contributions:** Conceptualization, L.L. and T.W.; methodology, L.L. and H.Z.; software, Y.Z. and H.W.; validation, T.W. and H.W.; formal analysis, Y.Z.; investigation, Y.Z. and H.W.; resources, X.H.; data curation, Y.Z.; writing—original draft preparation, L.L. and T.W.; writing—review and editing, L.L., X.H. and T.W.; visualization, Y.Z.; supervision, L.L., H.Z., X.H. and T.W.; project administration, L.L.; funding acquisition, L.L. and T.W.

**Funding:** This research was funded by National Natural Science Foundation of China (Grant No. 51778373), the Knowledge Innovation Project of Shenzhen (Grant No. JCYJ201703302143625006), Natural Science Foundation of SZU (Grant no. 082017), Natural Science Foundation Grant # CMMI 15-37431 and National Key Research and Development Program of China (2017YFB1201204).

**Acknowledgments:** The authors gratefully acknowledge K. Gurley of the University of Florida for providing the hurricane data and Lili Song of China Meteorological Administration for providing the typhoon data for this study.

**Conflicts of Interest:** The authors declare no conflict of interest. The funders had no role in the design of the study; in the collection, analyses, or interpretation of data; in the writing of the manuscript, or in the decision to publish the results.

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
