# Peer review of "An Analytical Framework for the Investigation of Tropical Cyclone Wind Characteristics over Different Measurement Conditions"

_applsci, doi:10.3390/app9245385_

Round 1

Reviewer 1 Report

In this study, the authors present an analytical model in order to analyze the turbulent wind characteristics for landfalling typhoons. The wind characteristics were then compared to seven well-known tropical cyclones. The content of this paper is significant because it can be used to examine design loads for buildings in regions that are directly impacted by landfalling storms. I've attached my line-by-line commentary to this document, but I wanted to comment on a few issues that can be used to strength this paper.

(1) The paper has a fairly large number of grammatical errors and typographical errors. I would suggest that the authors use an editor for these issues.

(2) The authors note the PDF of the hurricane integral scale shows a narrow distribution whereas the PDF of the typhoon integral scale has a significantly wider range. This is useful observation that is likely connected to the turbulent structure of these storms; however, no explanation was given for this observation. It would be useful to explain this in some detail.

(3) At end of the paper, the authors note that "... the AIJ-RLB-2004 makes a slightly higher estimation of the longitudinal turbulence intensity and a lower estimation of the longitudinal integral scale for both typhoons and hurricanes." A physically based explanation of this would be very useful.

Author Response

The authors truly appreciate the reviewer for his/her valuable suggestions and comments. We will address all your comments and suggestions in the attached file. Please check it, thank you!

Reviewer 2 Report

Review of “An analytical framework for investigation of tropical cyclone wind characteristics over different measurement conditions”

General comments:

This paper proposes an analytical framework for standardizing turbulent wind characteristics of tropical cyclones, including mean wind speed, turbulence intensity, integral scale, and gust factor. Then, actual wind observations are standardized by the framework to compare the differences between typhoon cases, hurricane cases, and previously recommended values. Although this reviewer cannot comment on the details of the proposed analytical framework based on some assumptions and empirical models, I have some concerns about the present manuscript from the perspective of describing tropical cyclones. Once these issues are addressed, I believe this manuscript to be published.

Recommendation: Minor revisions

Scientific comments:

The authors conclude that there are differences in standardized turbulent wind characteristics between typhoons and hurricanes. However, because data samples investigated are limited, the authors should not generalize that the characteristics differ between typhoons and hurricanes. Typhoons and hurricanes are essentially the same. The only difference is the naming. If there is regionality in the turbulent wind characteristics for tropical cyclones, the authors should investigate what causes the regionality. But I doubt it. More importantly, features of tropical cyclones vary depending on their sizes, intensities, inertial stability, quadrants relative to storm motion direction and vertical wind shear direction, radial locations relative to the radius of maximum wind, the magnitude of storm motion, land-sea contrast, and so on. Did the authors check the above properties and possible sampling biases in the datasets used? L333-335: The authors should not generalize the obtained results. L364-368: The authors should add “the observed” before “typhoons” and “hurricanes”. Figs. 5 and 6: Please explain what determines the upper limit of peak factors calculated. Are the reasons related to the difference between typhoons and hurricanes? L452: What is “diverse genesis mechanisms”? This paper does not investigate any genesis mechanisms. Why can the authors mention this possibility? And, how can the diversity of the genesis mechanisms affect turbulent wind characteristics?

Minor comments:

L135: Kepert (2006) does not deal with tropical cyclones in overland conditions. L148: Snaiki and Wu (2018) is not listed in references. L194, 197, and 199: What is the difference between “ln” and “log”?

Author Response

The authors truly appreciate the reviewer for your valuable suggestions and comments. We will address all your comments and suggestions in the attached file. Please check it, and thank you very much!
